# Temporary Consolidation of Marine Artifact Based on Polyvinyl Alcohol/Tannic Acid Reversible Hydrogel

**DOI:** 10.3390/polym15244621

**Published:** 2023-12-05

**Authors:** Qijun Huang, Jianrui Zha, Xiangna Han, Hao Wang

**Affiliations:** 1Institute of Cultural Heritage and History of Science and Technology, University of Science and Technology Beijing, Beijing 100083, China; huangqj1117@126.com (Q.H.); b2159283@ustb.edu.cn (J.Z.); 2National Center for Archaeology, Beijing 100013, China

**Keywords:** marine artifacts, temporary consolidation, extraction, mussel-like hydrogel, reversible removal

## Abstract

Underwater artefacts are vulnerable to damage and loss of archaeological information during the extraction process. To solve this problem, it is necessary to apply temporary consolidation materials to fix the position of marine artifacts. A cross-linked network hydrogel composed of polyvinyl alcohol (PVA), tannic acid (TA), borax, and calcium chloride has been created. Four hydrogels with varying concentrations of tannic acid were selected to evaluate the effect. The hydrogel exhibited exceptional strength, high adhesion, easy removal, and minimal residue. The PVA/TA hydrogel and epoxy resin were combined to extract waterlogged wooden artifacts and marine archaeological ceramics from a 0.4 m deep tank. This experiment demonstrates the feasibility of using hydrogel for the extraction of marine artifacts.

## 1. Introduction

The development of marine technology facilitates the detection of underwater archaeological artifacts [1]. A notable quantity of marine relics, such as shipwrecks, metal artifacts [2], lacquerware [3], ceramics [4,5], and jade, have been discovered. These archaeological relics record ancient society’s information of business, transport, and culture. To utilize marine relics, it is necessary to extract them from the ocean [6]. However, after an artifact is submerged, it becomes vulnerable to deterioration prompted by the marine environment. This includes the penetration and erosion of seawater salts into the pores of artifacts [7], the growth of algae, corals, and other organisms [8], and the corrosion of marine substrate. Due to the persistent impact of these factors, underwater cultural relics have become incredibly delicate and cannot be collected directly. Therefore, it is necessary to perform a temporary solidification process on underwater artifacts before extraction.

The SASMAP (Development of Tools and Techniques to Assess Stabilization Monitoring and Protection of Underwater Archaeological Sites) project, which is funded by the European Commission, has recently proposed two routes for the temporary consolidation of underwater artifacts. One approach employs using polymeric consolidation materials such as 3M™Scotchcast™Plus bandages, epoxy composites, and Carbomer gels. These materials possess the potential to conduct cross-linking in the underwater environment, resulting in the formation of a high-strength product that can provide support for fragile artifacts, enabling a solid-phase extraction. However, it has been observed that the curing strength of 3M bandage and epoxy resin is excessively high, which can lead to the formation of hollow drums and increase the danger of damaging the artifacts during lifting. Adding a complicated structural apparatus of carbomer gel can significantly decrease this risk [9]. The second option is the liquid nitrogen freezing method, which can extract cultural relics in an environmentally friendly process, but the effect lacks sufficient research [10]. Moreover, the Heritage Conservation Laboratory of Zhejiang University conducted research on two sublimated materials, veratraldehyde [11] and 4-dihydrochromanone [12], which had densities higher than that of water. These two materials are used to solidify underwater artifacts and combined with epoxy resin PU sponge [13] to extract several items, such as bead strings [14], ceramics [15], bamboo slips [16], and waterlogged wooden artifacts [17]. The study of Veratraldehyde and 4-dihydrochromanone as temporary solidification materials for underwater cultural relics aims to find new routes. However, the current packaging method is highly reliant on the underwater environment and diver operation.

The temporary consolidation of underwater cultural relics needs the following three requirements. The first is to guarantee the safety of artifacts. Due to the ocean environment, the cultural relics have become fragile. Therefore, high-strength solid materials are needed to shield these fragile artifacts from the impact of water currents and tidal action. The second factor is convenience. Each member of the underwater archaeology staff is allocated a dive duration of 30–40 min. The underwater environment of a shipwreck is characterized by poor visibility and may be exceedingly challenging. The third is to keep the integrity of artifacts. Ceramics are arranged in a certain pattern that can reflect the ancient ship life. To enhance the preservation of archaeological information, it is necessary to consolidate the artifacts and then extract them as a cohesive unit. So, ensuring the security of artifacts, guaranteeing archaeological information, and simplifying underwater operations have emerged as major concerns.

An ideal material for extracting fragile underwater artifacts would be a consolidate that exhibits wet adherence properties and can be easily removed after extraction due to its reversible adhesion capabilities. Hydrogel is a polymeric material with a three-dimensional network structure, exhibiting characteristics of both solids and liquids. The extensive range of characteristics and adaptability have resulted in its utilization in several applications in the fields of biomedicine and material science. Various cross-linking methods provide hydrogels with distinct mechanical and reversible properties. The reversible hydrogels network is formed by non-covalent bonding, such as hydrogen bonding, ionic bonding, and hydrophobic interaction [18], as well as dynamic covalent bonding including imine bonding, disulfide bonding, and Diels–Alder reactions [19]. This hydrogel exhibits the capacity to respond to external stimuli, mainly affected by hydrogel adsorption properties [20], swelling properties [21], and self-healing ability [22]. Hydrogels were introduced to the field of conservation in the late 1980s. Wolbers initially used polyacrylic acid as a gelling agent to eliminate dust and coatings from the surface of oil paintings [23]. Subsequently, hydrogels were used for the purpose of cleaning stone surfaces [24], metal [25], frescoes [26], and other artifacts. The sol–gel transition is a significant characteristic of hydrogels, allowing the formation of bonding interfaces with different materials [27]. This property enables the production of gels that stick effectively to surfaces, providing the potential to use this kind of material to consolidate the artifacts. Guo used poly vinyl alcohol (PVA) and polyacrylamide to form an interpenetrating network structure. The precursors of these polymers were able to permeate the micropores and establish strong hydrogen bonds with the surface of the fragments [28]. The hydrogel has strong adhesion properties even in moist conditions and effectively adheres to ceramic surfaces. Hydrogel has demonstrated its efficacy as a secure, environmentally friendly, and adaptable substance in several cultural heritage preservation fields. It has great potential to be applied in the field of the temporary fixation of underwater cultural relics. Hydrogels with both reversibility and adhesion can effectively solve the problem of information retention during the extraction of underwater artifacts.

PVA hydrogel possesses exceptional mechanical properties, such as remarkable strength and elasticity. Moreover, it can be crosslinked with multifunctional groups, which facilitates the formation of adhesion with the interface. However, obtaining interfacial adhesion in an aqueous medium is more difficult than achieving the same level of interfacial adhesion in an air environment. The formation of a hydration layer on the underwater artifact’s surface hinders the interaction between adhesive and artifacts [29]. Inspired by research on bio-condensates of underwater artifacts [30] and marine antifouling [31], mussel-like hydrogels have been developed. A catechol structure was introduced into the PVA borax hydrogel to improve the underwater adhesion properties and to obtain reversible properties. Tannic acid is a plant-derived polyphenolic molecule that possesses a catechol structure and can react with a variety of functional groups, resulting in adhesion and reversibility. So, tannic acid and Ca^2+^ ions were added to enhance the capabilities of a PVA-borax hydrogel. The analysis result shows that modified hydrogel exhibited remarkable adhesion qualities in underwater environments and could be easily removed without causing damage.

## 2. Materials and Methods

### 2.1. Materials

PVA (average Mw ≈ 1750, 97% hydrolyzed), tannin acid (TA) (F.W. = 1701.02, >92%), borax (F.W. = 381.37, AR), CaCl_2_ (F.W. = 110.98, AR), and disodium ethylenediaminetetraacetic acid (Na_2_EDTA) (F.W. = 372.24, AR) were obtained from Sinopharm Chemical Reagent Co., Ltd. (Shanghai, China). Water was purified using a JRO-EDI-P19 gradient system.

The waterlogged woods were obtained from the (Nanhai NO. I shipwreck, Yangjiang, Guangdong province, China) and (Zhiyuan shipwreck, Qingdao, Shandong province, China). The marine archaeological ceramics were obtained from the (Shengbei yu, Zhangzhou, Fujian province, China) (Yuan dynasty). Samples were supplied by the National Center for archaeology.

### 2.2. Methods

Hydrogels with different tannic acid concentrations were prepared and tested (Table 1). First, different concentrations of aqueous solutions were obtained by adding appropriate amounts of PVA and TA powders to distilled water heated at 90 °C for 1 h with mechanical stirring. To the aqueous solution, 10 mL of borax solution (0.4 g/L) and 5 mL of calcium chloride (0.08 g/L) solution were added.

### 2.3. Preparation of Concretion

Inspired by the hydrogel cleaning calcium mockup [32], a calcium carbonate model was prepared to assess the adhesion properties of PVA/TA hydrogel. Calcium powder was poured into a metal mold (2 cm × 2 cm × 3 cm) and held at a pressure of 3.5 MPa for 1 min. It was fired at 500 °C for 1 h with a temperature increase rate of 5 °C/min.

Pottery mockups were prepared to assess the response properties of PVA/TA hydrogels. The pottery powder was poured into a metal mold (2 cm × 2 cm × 3 cm) and held at a pressure of 3.5 MPa for 2 min. It was fired at 1100 °C for 1 h with a temperature increase rate of 3.6 °C/min.

### 2.4. Fourier Transform Infrared Spectrometer

The hydrogels were characterized using a Fourier infrared spectrometer (Thermo Fisher Nicolet iS 5, Massachusetts, America). The hydrogel samples were freeze-dried, mixed, and pressed with KBr, and FR-IT spectra were recorded in the wave number range of 400–4000 cm^−1^ with a scanning resolution of 16 cm^−1^.

A Fourier infrared spectrometer was used to characterize the hydrogels before and after oxidation. The scanning range was 1200–2000 cm^−1^, and the scanning resolution was 16 cm^−1^.

Infrared spectroscopy was performed using Fourier infrared spectroscopy on the Na_2_EDTA stock solution and the leach solution coated on KBr window slices. The scanning range was 400–4000 cm^−1^ with a scanning resolution of 16 cm^−1^.

### 2.5. Scanning Electron Microscope

Scanning electron microscope images of the hydrogels were obtained using a Regulus 8100 cold-field scanning electron microscope, and hydrogel pore sizes were counted using Image J ((version 1.54f). The hydrogels were freeze-dried, and liquid nitrogen was embrittled for gold spraying.

### 2.6. Adhesion Performance

Shells, waterlogged woods, iron sheets, and ceramic mockups were selected as objects and placed in a 3.6% saline environment, and hydrogels were adhered to the surface and gently pressed for 10 s to observe the adhesion of hydrogels to different substrates.

The hydrogel was adhered to the surface of the calcium mockup samples and was separated after gently pressing for 10 s, and the changes in the mass of the calcium mockup samples were recorded [33].

### 2.7. Rheological Measurements

Rheological measurements of hydrogels were tested using an Anton Paar MCR 302 rheometer. The linear viscoelastic range of the hydrogels was measured at a shear frequency of 10 Hz in the range of 0.1–100 rad/s. The constant stress was 1% and the energy storage modulus and loss modulus of the samples were measured in the frequency range of 0.1–100 rad/s. The frequency was set to 1 Hz, and the rheological properties of the hydrogel were measured by applying high and low varying stresses to the probe for a certain period. The program was set as follows: 120 s, 1% low strain; 120 s, 300 high strains; and 300 s, 1% low strain.

### 2.8. Reversible Performance

Hydrogels have oxidation and acid environment response properties that can be exploited for the reversible removal of hydrogels. To study the oxidation response properties of hydrogels, hydrogels were placed in an open environment. The hydrogels’ color and fluorescence variations under 365 nm UV light were documented. 

To study the responsiveness of the hydrogels in acid or weak acid environments, the hydrogels were soaked in 10 mmol Na_2_EDTA solution, and the hydrogel morphology changes were recorded.

### 2.9. Simulation Application

A simulation application was conducted to extract underwater artifacts utilizing a traditional epoxy multilayer material combined with PVA/TA hydrogel. A brush was used to clean the sediment near the artifacts, and the visible portion of the artifacts was covered with PVA/TA hydrogel. The artifact was fortified by inserting a steel plate at its base and enveloping it with multilayer epoxy material. Additionally, steel strips were applied around the edges to prove reinforcement. After the completion of the curing process of the epoxy resin, the artifacts can be extracted. Once the artifacts have been extracted from water, the PVA/TA hydrogel on the surface of the artifacts can be removed using an oxidation method or soaking in Na_2_EDTA. Figure 1 illustrates the extraction procedure.

## 3. Results and Discussion

### 3.1. Hydrogel Characterization

To evaluate the effectiveness of introducing TA addition, FT-IR spectroscopy was used to identify any changes in the functional group within the gel. The infrared spectra of PVA, PVA-Borax, PVA-TA-Borax, and PVA-TA-Borax-CaCl_2_ are shown in Figure 2. The gel composite of PVA and borax exhibited distinct peaks of boric acid and borate at 1341 cm^−1^ and 841 cm^−1^ [34]. Tannic acid has numerous catechol structures. Catechol exhibits hydrogen bonding with PVA, resulting in broadening and downshifting of the original O-H peak (3444 cm^−1^). The phenolic hydroxyl group is characterized by a peak at 1467 cm^−1^. The catechol structure is a special functional group that possesses adhesion properties. Upon the addition of Ca^2+^, it splits into two distinct peaks (1472 cm^−1^ and 1462 cm^−1^), suggesting the successful formation of a complexation with tannic acid [35]. The PVA/TA hydrogel has borate bonds, hydrogen bonds, and metal–phenol coordination bonds. The multi-crosslinked network of the hydrogel provides a high strength that helps consolidate the artifacts.

A homogeneous hydrogel structure is a benefit to ensuring the successful extraction of submerged artifacts. SEM images can effectively detect the morphology and pore distribution of hydrogels. Figure 3 shows the SEM image of PVA/TA hydrogel. The PVA/TA hydrogel has a distinct porosity structure like a sponge. These holes are uniformly distributed inside the hydrogel and in conjunction with the air bubbles formed during the preparation. The presence of tannic acid will impact the microstructure and pore size of the hydrogel. The pores of the PVA/TA0.7 hydrogel exhibited considerable changes, resulting in the formation of a less dense network structure and an enlargement of the pore size in the hydrogel. Tannic acid generated numerous hydrogen bonds with PVA, therefore inhibiting the formation of chemical bonds.

Using Image J software to analyze the scanning electron microscope images of the hydrogel, it is possible to obtain a statistical map illustrating the pore size distribution of the hydrogel, as shown in Figure 3e. The PVA/TA hydrogel exhibited a nanometer scale pore size, ranging from 5 to 100 nm, with many pores centered in the 5–10 nm region. The pore size of the PVA/TA0.7 hydrogel exhibited a significant increase, possibly because the density of the crosslinked network decreased because of enhanced hydrogen bonding. 

The mechanical properties of hydrogels have a direct impact on their ability to provide protection and residuals. The rheological test is effective in assessing these features. Figure 4a illustrates the stress scanning curves of PVA/TA hydrogels with varying amounts of tannic acid. It is observed that within the low-stress range (0.01~1%), the hydrogel’s energy storage modulus gradually increases with the applied stress increase. In the linear viscoelastic range of the hydrogel, the energy storage modulus reaches the maximum and tends to stabilize. The PVA/TA0.5 hydrogel has the highest energy storage modulus, and the internal bond of the hydrogel is saturated. Figure 4b exhibits the frequency scan curves of each group of PVA/TA hydrogels. The frequency scan curves of the hydrogels have a quadratic function shape. This result indicates that the PVA/TA hydrogels conduct dynamic crosslinking [36,37]. Applying pressure to the hydrogel can effectively demonstrate the hydrogel’s ability to provide cushioning (Figure 4c–f). The PVA/TA0.5 hydrogel possesses the highest resistance to external pressures among all the hydrogels.

The PVA/TA hydrogel exhibited borate bonds, metal–phenol coordination bonds, and hydrogen bonds. External pressures can cause the hydrogel’s dynamic bonds to reconstruct, leading to self-healing. The rheometer results of the hydrogel are shown in Figure 5. The hydrogel demonstrated a gel–sol–gel transition in response to changes in stress. The PVA/TA0.4 restoration reached 91.6% of its initial condition after 5 min, demonstrating a remarkable self-repairing capability. The PVA/TA0.5 sol–gel exhibits a transition time of 33 s and is capable of returning to its initial condition within 132 s. The hydrogel’s network can be completely restored by dynamic cross-linking bonds. Hydrogels exhibit exceptional flexibility and the ability to self-heal, making them ideal for safeguarding artifacts of various forms during extraction and transportation.

### 3.2. Application Properties

The hydrogel exhibits wet adhesion ability due to the presence of catechol. In order to assess the efficacy of hydrogel on various types of underwater artifacts with different surface properties, it is important to conduct evaluations. Figure 6 demonstrates the ability of PVA/TA hydrogel to bond with different substrates, such as shell, iron sheet, pottery sheet, and waterlogged wood, in a simulated ocean environment. These substrates are representative of common artifacts found underwater. 

Catechol is prone to oxidation in the atmosphere, leading to a decrease in the hydrogel’s adhesion. Figure 7c illustrates a gradual decrease in the adhesive properties of the PVA/TA hydrogel as time progresses. The hydrogel exhibited a dynamic color alteration (Figure 7a), changing from light yellow to dark brown. The fluorescence of the hydrogel was quenched (Figure 7b). The hypoxic conditions in the underwater environment might impede the degradation and detachment of the hydrogel throughout its use.

FT-IR was used to explore the adhesion decrease process of the hydrogels (Figure 8). The telescopic vibration peak of the benzene ring in the pristine hydrogel is observed at 1648 cm^−1^. This peak has a blue-shift to 1630 cm^−1^. This shift is attributed to the conjugation between the C=O group and the aromatic ring, resulting in a decrease in the frequency of the characteristic peak of the benzene ring. Additionally, the catechol structure undergoes oxidation to form benzoquinone [38]. In addition, the characteristic peaks corresponding to the metal–phenol coordination bonds in the hydrogel (at 1472 cm^−1^, 1462 cm^−1^) and the characteristic peak of the borate bond at 1341 cm^−1^ remained unchanged. Only catechol underwent oxidation to yield benzoquinone within the hydrogel. Thus, by utilizing the hydrogel’s oxidation response property, it becomes feasible to eliminate the hydrogel from the surface of extracted underwater artifacts. This property fulfills the requirements of temporary underwater consolidation materials.

The PVA/TA hydrogel network consists of borate bonds, metal–phenol coordination bonds, and hydrogen bonds. Borate bonds and metal–phenol coordination bonds exhibit high sensitivity to fluctuations in pH value. The tests employed a Na_2_EDTA solution to provide an acidic environment to eliminate the hydrogel. After a period of 48 h, the hydrogel exhibits a transformation from a yellowish gel state to a macromolecular sol. The dissolution of tannic acid in the hydrogel resulted in the solution gaining a light yellow color (Figure 9a). The Na_2_EDTA stock solution and the leach solution were characterized via FT-IR spectroscopy. The results are presented in Figure 9b. The Na_2_EDTA solution exhibited antisymmetric stretching vibration peaks and symmetric stretching vibration peaks of the carboxyl group of COO-Na at 1637 cm^−1^ and 1489 cm^−1^. Part of the EDTA present in the leaching solution formed complexes with the calcium ions in the hydrogel, resulting in the appearance of a symmetric stretching vibration peak at 1404 cm^−1^ [39]. The B-O bond in the EDTA compound exhibited a distinct peak at a wavenumber of 841 cm^−1^. The PVA/TA hydrogel displaced the metal–phenol coordination bond and borate ester bond in the Na_2_EDTA solution, leading to the disintegration of the hydrogel and its reversion to a sol state. Therefore, immersing extracted underwater artifacts in Na_2_EDTA solution can effectively remove surface hydrogel.

The application test was conducted on waterlogged wood specimens retrieved from the shipwreck of the “Nanhai NO. I”. The waterlogged wood samples had a moisture content of around 1180%, indicating significant spoilage. The waterlogged wood was immersed in deionized water for the purpose of oxidation. After 48 h, the hydrogel suffered oxidation and became detached from the wood’s surface. Tiny alterations in the morphology of the wood samples were detected (Figure 10a). The SEM-EDS analysis results of the detached wood samples revealed that the hydrogel had no impact on the wood samples (Appendix A). The EDS data indicate (Appendix A) a minute amount of B and Ca present on the surface of the wood sample. The used hydrogel’s surface was quantified based on the characteristic peak intensity of the borate bond (1341 cm^−1^) (Figure 10g). The blue region of the PVA/TA0.5 sample was similar to the untreated sample surface, suggesting that the hydrogel had a lower concentration of borate bond residue compared to the other groups.

The hydrogel was removed from the archaeological wood sample’s surface by immersing it in Na_2_EDTA solution. After a duration of 48 h, the hydrogel was successfully removed from the wood samples without altering the original shape of the wood artifacts. The SEM-EDS results indicate (Appendix A) that the B content on the wood surface after hydrogel removal remained almost unchanged when the hydrogels were removed. The cause of this phenomenon was the complete disintegration of the hydrogels in Na_2_EDTA solution, while the PVA/TA0.6 hydrogels did not undergo a full reaction (Appendix A). Combined with the micro infrared data (Figure 11), it can be observed that the PVA/TA0.5 wood sample exhibited the most extensive blue region and the lowest amount of borate bond residue after the removal of the hydrogel. On the other hand, the PVA/TA0.6 wood sample exhibited a noticeable red area even after the hydrogel was removed. Comparing the above methods, it is evident that immersing the waterlogged woods in Na_2_EDTA solution is a more effective approach.

The pre-wetted ceramic mockup samples were exposed to the air environment to facilitate oxidization and the removal of the hydrogels. The PVA/TA hydrogel was removed from the block’s surface by air oxidation, and this method did not affect the morphology of the block samples (Figure 12). The SEM-EDS (Appendix A) results indicate that the PVA/TA0.4, PVA/TA0.5, PVA/TA0.6, and PVA/TA0.7 samples showed an increase in the content of B and Ca, as well as a reduction in the content of Fe. The rise in the B and Ca contents can be attributed to the presence of hydrogel residue on the ceramic mockup samples. The decrease in Fe can be attributed to the coordination reaction between the catechol molecule and Fe. The ceramic mocked samples treated with the PVA/TA0.4 and PVA/TA0.5 groups exhibited negligible elemental alterations. The PVA/TA0.6 and PVA/TA0.7 hydrogels-treated samples resulted in extensive alterations of Fe, and the color of the ceramic mocked samples was changed. More information could be obtained from 2D-IR analysis conducted at the frequency of 1341 cm^−1^ for the pottery samples after the removal of hydrogel (Figure 12). The PVA/TA0.4 and PVA/TA0.5 samples exhibited a comparable blue area to the control sample while also displaying a reduced presence of borate bond residues. 

The hydrogels adhered to the surface of the pottery samples were removed by immersion in a Na_2_EDTA solution. After 48 h, the hydrogels composed of PVA/TA0.4 and PVA/TA0.5 disintegrated, and the morphology of the ceramic blocks remained unchanged. However, the PVA/TA0.6 and PVA/TA0.7 samples exhibited the presence of black residues on their surfaces (Figure 13a). An SEM-EDS analysis was conducted on the black residues (Figure 13b), and the corresponding EDS findings are included in Appendix A. The black substance should consist of a hydrogel complexed with iron ions. The presence of Na_2_EDTA solution induced the Fe^3+^ ions in the mockup samples aggregated on the surface and bind with the hydrogel catechol, resulting in the formation of black deposits. The ceramic samples were assessed using SEM-EDS after eliminating the black deposits. The SEM-EDS (Appendix A) results indicate that the four groups of samples exhibited a significant increase in the B element and a decrease in Ca and Fe.

The presence of hydrogel residue on the surface of the pottery samples can be visualized by 2D-IR imaging (Figure 13). The absence of black residues in PVA/TA0.4 and PVA/TA0.5 can be attributed to the low catechol concentration in the hydrogel. The FT-IR maps of PVA/TA0.6 and PVA/TA0.7 exhibited a large region of red color. Therefore, it is assumed that the remaining substance is a PVA/TA hydrogel that is extensively cross-linked and bound with Fe^3+^. Consequently, when dealing with pottery, the hydrogel cannot be removed by immersing it in Na_2_EDTA or a weak acid solution. Instead, it should be removed by oxidation. The PVA/TA0.5 hydrogel exhibited the lowest residual content after removal.

The marine archaeological ceramics which originate from the Shengbeiyu shipwreck, Yuan Dynasty, Fujian Province, were buried in a tank with a sand bottom that was 0.4 m deep. The laboratory conducted simulations of ceramic extraction employing a combination of PVA/TA0.5 hydrogel and epoxy resin multilayer material, as shown in Figure 14. The hydrogel was inserted between the epoxy multilayer material and the artifacts to avoid the artifacts being displaced because of the hollowing of the epoxy multilayer material. Following the extracted process, the hydrogel was removed by oxidation. After the removal of the hydrogel, the position of the porcelain bowls remained unchanged, and there was no hydrogel residual present on the artifacts. The waterlogged wooden artifacts originated from the Zhiyuan shipwreck in Shandong Province and were degraded specimens. The wooden artifacts were randomly placed and buried in a water tank with a sand bottom that was 0.4 m deep. Simulation experiments were conducted using a PVA/TA0.5 hydrogel combined with an epoxy resin multilayer material (Figure 15). After the inclusions were safely extracted from the water, the hydrogel was eliminated by immersing it in a 10 mmol Na_2_EDTA solution. After hydrogel removal, the wood artifact arrangement position and morphology remain unchanged. The underwater extraction process was completed within 20 min and meets the requirements of underwater archaeological excavation.

## 4. Conclusions

To address particular issues, such as location information errors and damage to cultural relics during the extraction of underwater artifacts, PVA/TA hydrogel was prepared. PVA/TA hydrogel is a substance that provides cushioning and has an adhesion effect. It can be used to consolidate artifacts together in an underwater environment and fix the position of artifacts. The hydrogel can be easily removed by oxidation or immersing in a mild acid solution. The analysis results demonstrated the formation of a compact network structure of PVA/TA0.5 hydrogel. This hydrogel exhibited excellent wet adhesion properties with a rheological strength of 12,000 Pa. Moreover, the hydrogel can be efficiently removed by air oxidation and weak acid solution without residue in the wood and pottery. The combination of PVA/TA hydrogel and epoxy resin multilayers offers a straightforward and efficient solution for extracting artifacts. This method is particularly advantageous for underwater operations due to its simplicity of use.

## Figures and Tables

**Figure 1 polymers-15-04621-f001:**
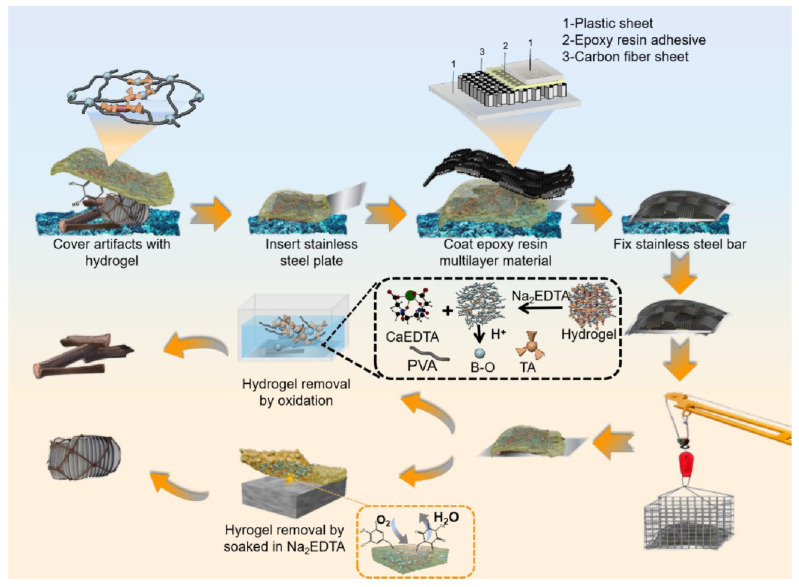
The extraction process of underwater artifacts with hydrogel temporary.

**Figure 2 polymers-15-04621-f002:**
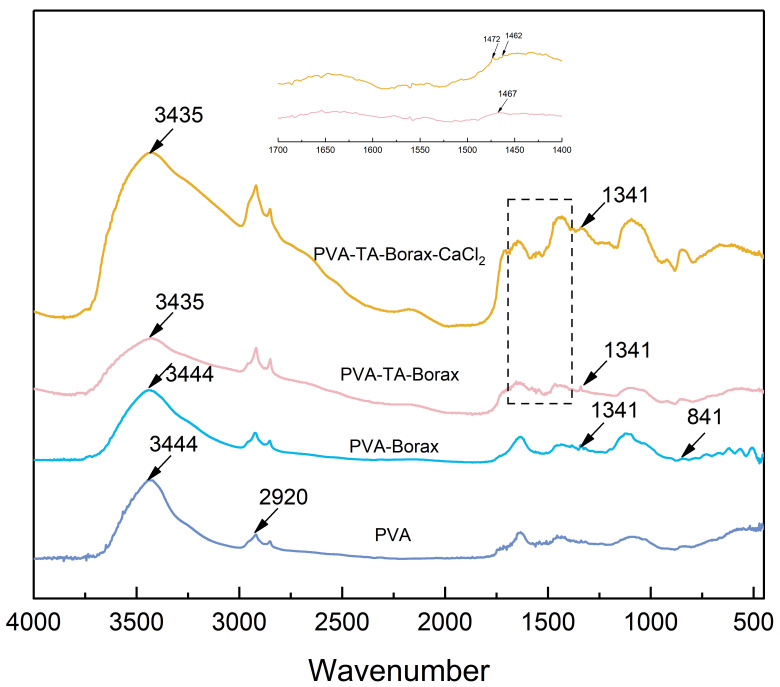
FT-IR spectra of the powder (PVA) and hydrogel (PVA-Borax, PVA-TA-Borax, PVA-TA-Borax-CaCl_2_).

**Figure 3 polymers-15-04621-f003:**
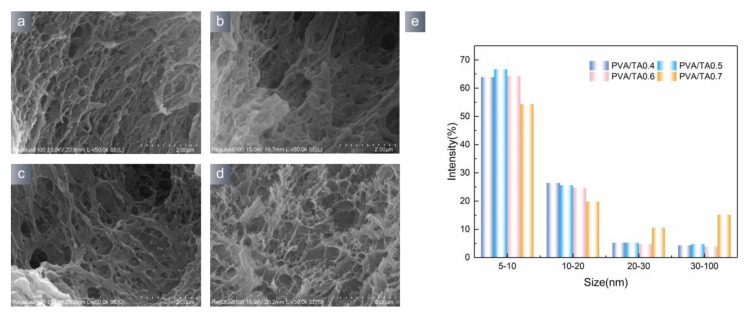
SEM image of hydrogel: (**a**) PVA/TA0.4; (**b**) PVA/TA0.5; (**c**) PVA/TA0.6; (**d**) PVA/TA0.7; (**e**) pore size distribution of hydrogel.

**Figure 4 polymers-15-04621-f004:**
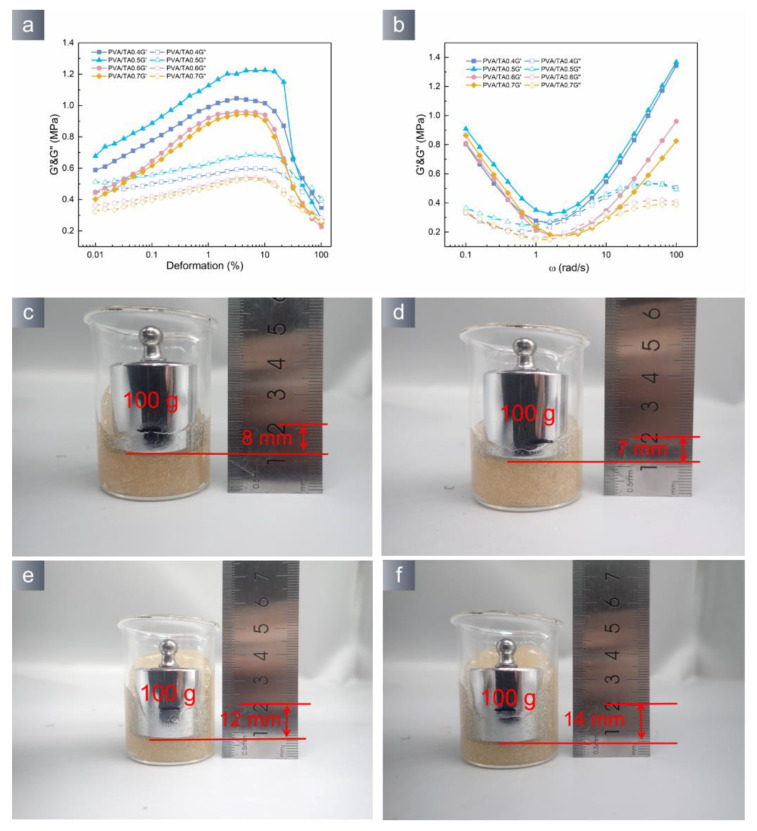
(**a**) Amplitude sweep and (**b**) frequency sweep curves of hydrogel; external forces can deform the hydrogel (**c**–**f**).

**Figure 5 polymers-15-04621-f005:**
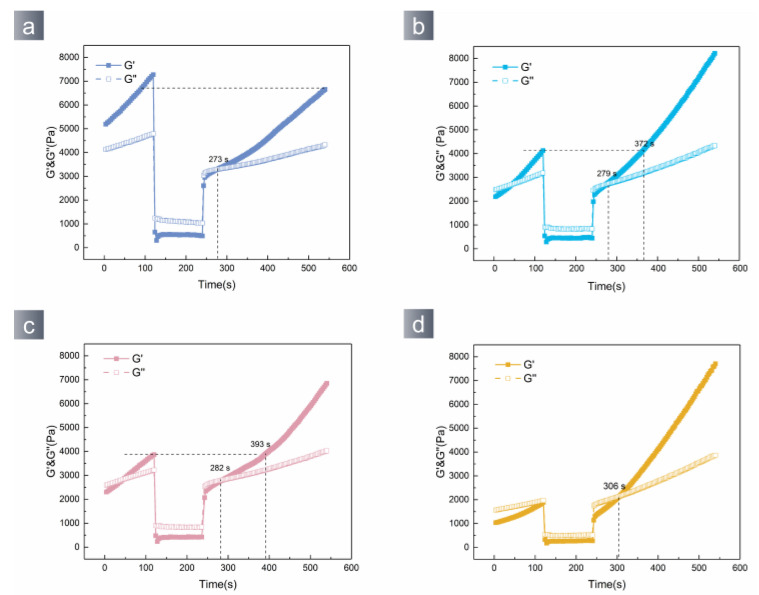
Amplitude changing sweep of hydrogel: (**a**) PVA/TA0.4, (**b**) PVA/TA0.5, (**c**) PVA/TA0.6, (**d**) PVA/TA0.7.

**Figure 6 polymers-15-04621-f006:**
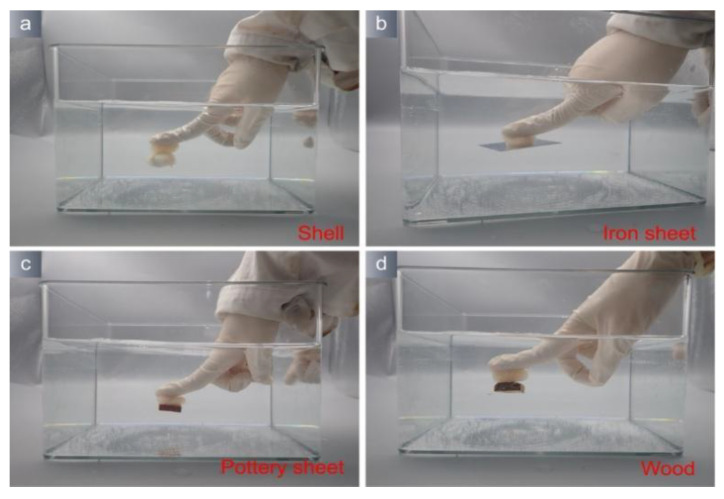
Hydrogels adhere to various substrates. (**a**) Hydrogel adhere to shell; (**b**) Hydrogel adhere to iron sheet; (**c**) Hydrogel adhere to pottery sheet; (**d**) Hydrogel adhere to wood.

**Figure 7 polymers-15-04621-f007:**
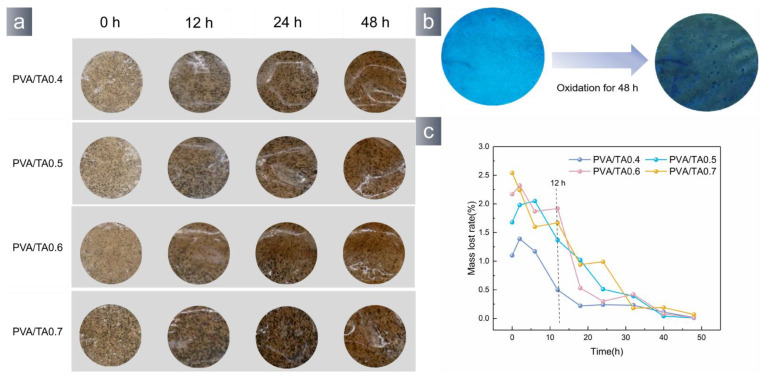
(**a**) Color changes during oxidation of hydrogels, (**b**) fluorescent quenching, (**c**) The adhesion strength decreases during oxidation.

**Figure 8 polymers-15-04621-f008:**
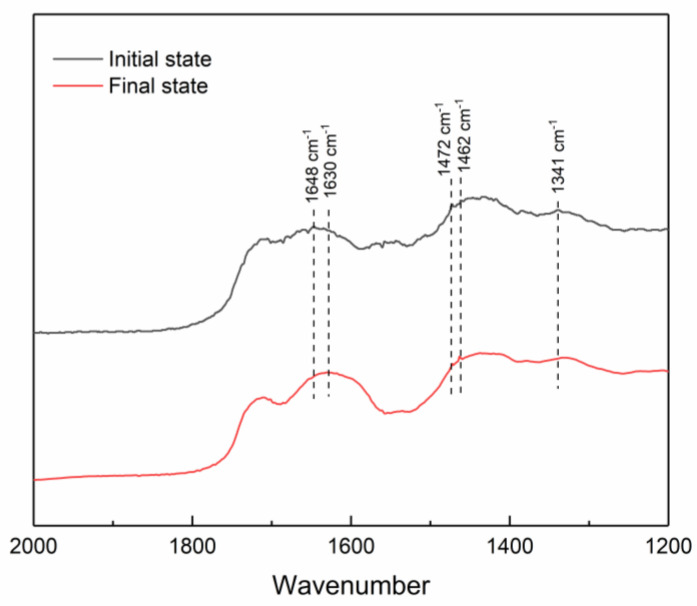
FT-IR spectra of the hydrogel oxidation process.

**Figure 9 polymers-15-04621-f009:**
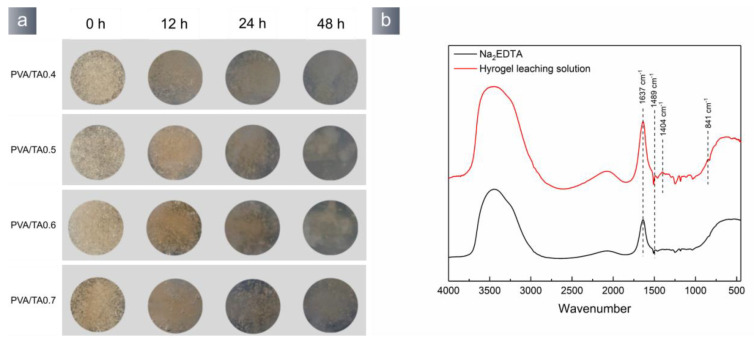
(**a**) The process of PVA/TA hydrogel removal by immersing in 10 mmol Na_2_EDTA solution; (**b**) FT-IR of Na_2_EDTA before and after hydrogel immersion.

**Figure 10 polymers-15-04621-f010:**
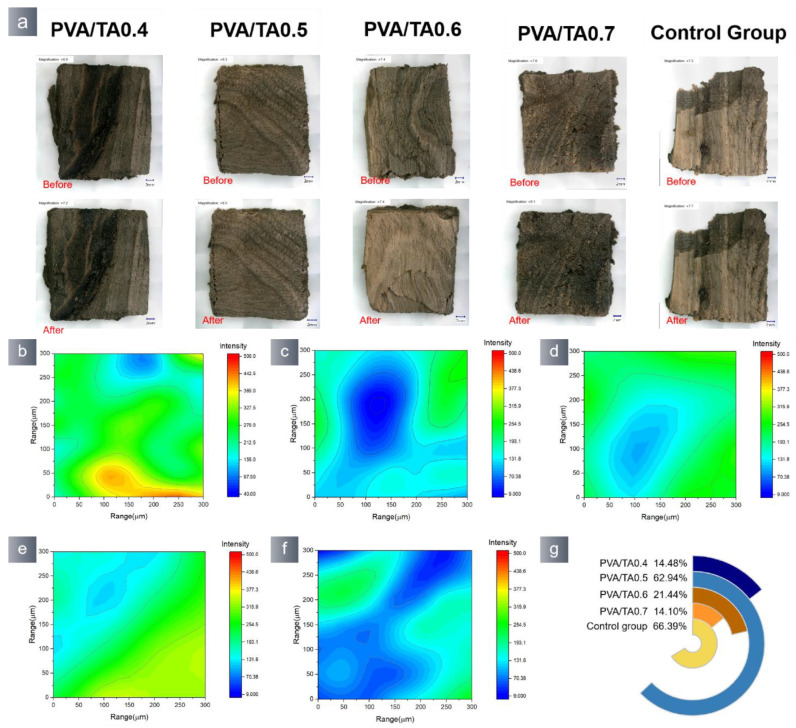
(**a**) Morphological changes in wood before and hydrogel removal by oxidation. Used wood maps obtained from the 1341 spectra that make up the FTIR map: (**b**) PVA/TA0.4; (**c**) PVA/TA0.5; (**d**) PVA/TA0.6; (**e**) PVA/TA0.7; (**f**) control group, and (**g**) measuring the blue area in each FTIR map.

**Figure 11 polymers-15-04621-f011:**
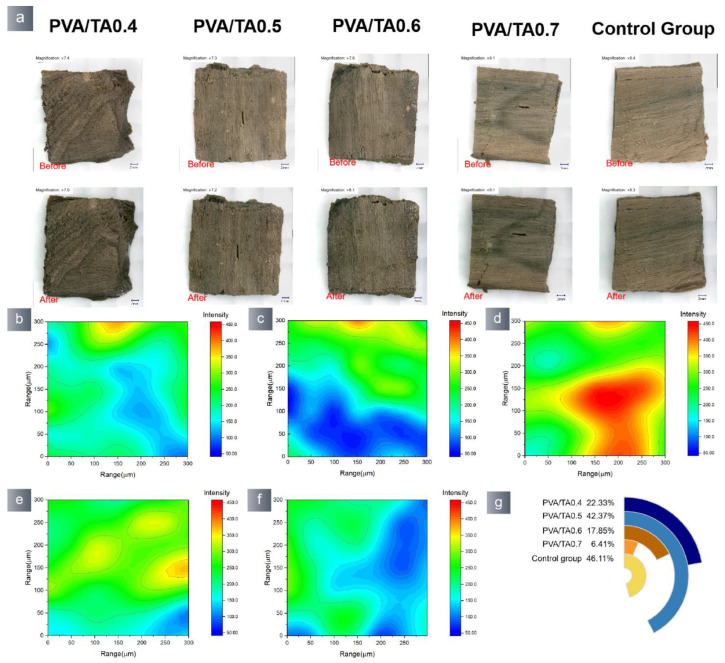
(**a**) Morphological changes in wood after hydrogel removal by soaking in 10 mmol Na_2_EDTA solution. Used wood maps obtained from the 1341 spectra that make up the FTIR map: (**b**) PVA/TA0.4; (**c**) PVA/TA0.5; (**d**) PVA/TA0.6; (**e**) PVA/TA0.7; (**f**) control group, and (**g**) measuring the blue area in each FTIR map.

**Figure 12 polymers-15-04621-f012:**
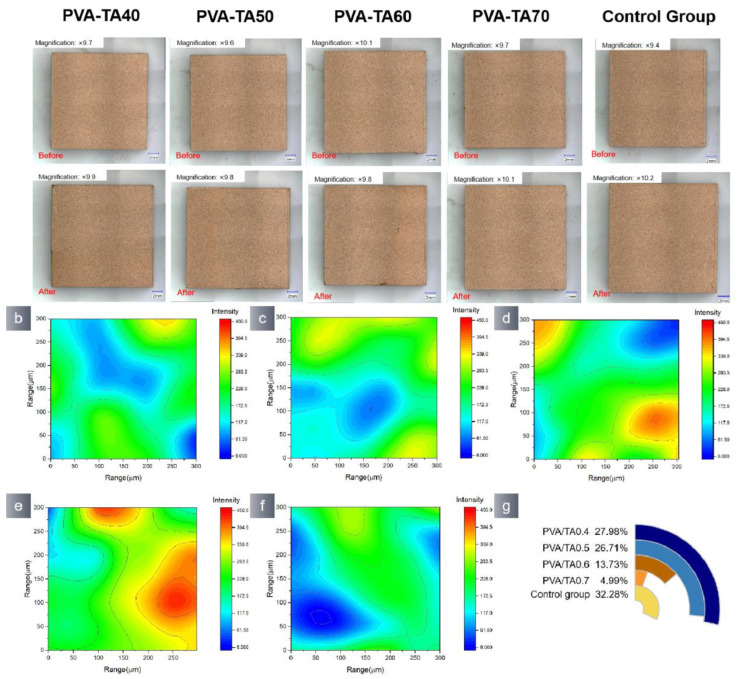
(**a**) Morphological changes in pottery before and hydrogel removal by oxidation used pottery maps obtained from the 1341 spectra that make up the FTIR map: (**b**) PVA/TA0.4; (**c**) PVA/TA0.5; (**d**) PVA/TA0.6; (**e**) PVA/TA0.7; (**f**) control group, and (**g**) measuring the blue area in each FTIR map.

**Figure 13 polymers-15-04621-f013:**
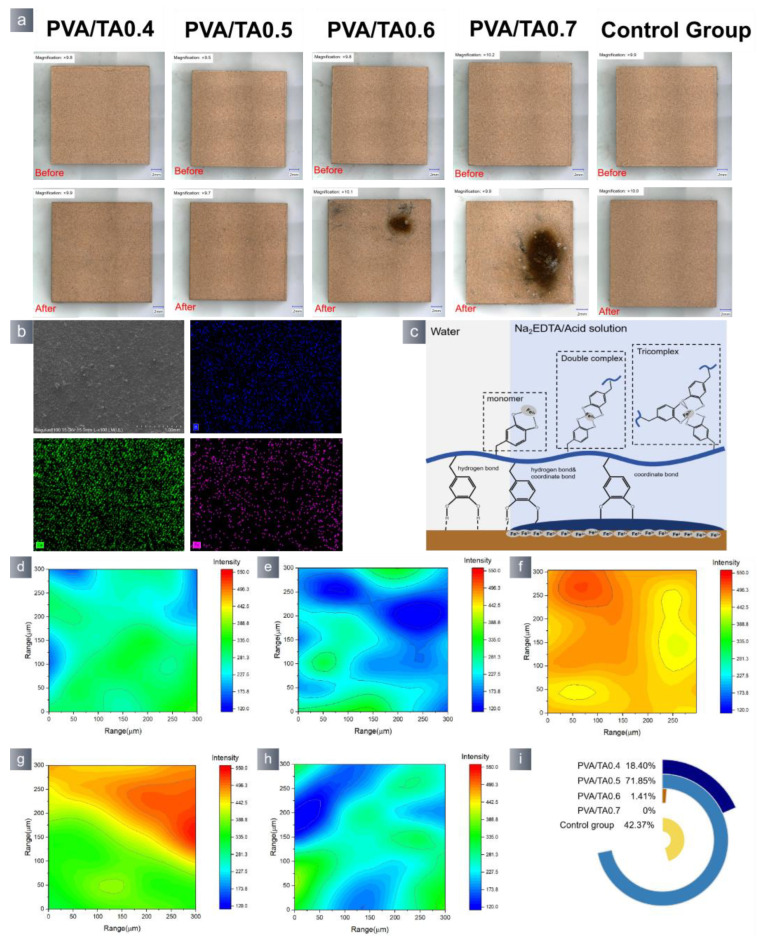
(**a**) Morphological changes in pottery before and hydrogel removal by soaking in 10 mmol Na_2_EDTA solution. (**b**) The SEM-EDS sweep of black sediment. (**c**) Black sediment formation mechanism used pottery maps obtained from the 1341 spectra that make up the FTIR map: (**d**) PVA/TA0.4; (**e**) PVA/TA0.5; (**f**) PVA/TA0.6; (**g**) PVA/TA0.7; (**h**) control group, and (**i**) measuring the blue area in each FTIR map.

**Figure 14 polymers-15-04621-f014:**
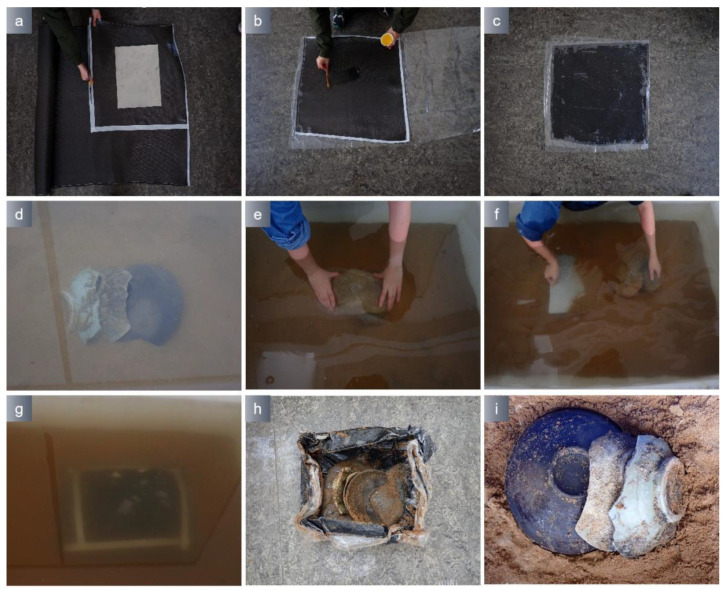
Hydrogel and epoxy resin multilayer material were used to extract porcelain (14 °C). (**a**) Cutted carbon fiber sheet; (**b**) Applied the epoxy resin to the carbon fiber sheet; (**c**) Epoxy resin multilayer material; (**d**) Buried marine ceramics in water; (**e**) Adhered hydrogel to marine ceramics; (**f**) Inserted the base plate; (**g**) Wrapped epoxy multilayer materials; (**h**) Removed the base plate; (**i**) Marine ceramics after hydrogel removal.

**Figure 15 polymers-15-04621-f015:**
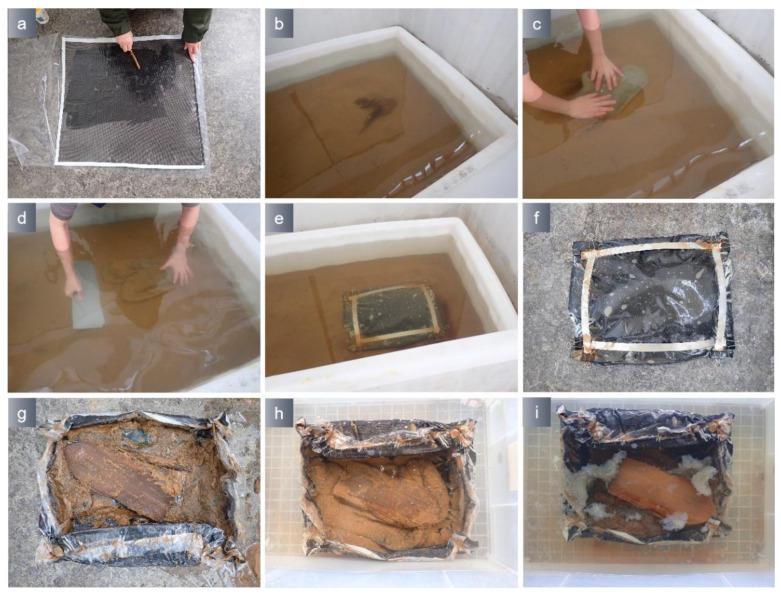
Hydrogel and epoxy resin multilayer material were used to extract wood (14 °C). (**a**) Applied the epoxy resin to the carbon fiber sheet; (**b**) Buried waterlogged wooden artifacts in water; (**c**) Adhered hydrogel to waterlogged wooden artifacts; (**d**) Inserted the base plate; (**e**) Wrapped epoxy multilayer materials; (**f**) Lifted the package out of the water (**g**) Removed the base plate; (**h**) soaked in Na_2_EDTA to remove hydrogel; (**i**) waterlogged wooden artifacts after hydrogel removal.

**Table 1 polymers-15-04621-t001:** Composition of the PVA/TA hydrogel.

Number	PVA (g)	TA (g)	H_2_O (mL)	Borax (mL)	CaCl_2_ (mL)
PVA/TA0.4	12	0.4	73	10	5
PVA/TA0.5	12	0.5	73	10	5
PVA/TA0.6	12	0.6	73	10	5
PVA/TA0.7	12	0.7	73	10	5

## Data Availability

Data are contained within the article.

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
