# Peer review of "Temporary Consolidation of Marine Artifact Based on Polyvinyl Alcohol/Tannic Acid Reversible Hydrogel"

_polymers, 2023, doi:10.3390/polym15244621_

Round 1
Reviewer 1 Report
Comments and Suggestions for Authors
Dear Authors
Underwater artifacts are vulnerable to damage and loss of archaeological information during the extraction process. To solve this problem, it is necessary to apply temporary consolidation materials to fix the location of marine artifacts.
The authors developed a cross-linked network hydrogel composed of polyvinyl alcohol, tannic acid, borax, and calcium chloride for this purpose. Four hydrogels with varying concentrations of tannic acid were selected to evaluate the effect. The hydrogel exhibited exceptional strength, high adhesion, a variety of removed methods, and minimal residue. The PVA/TA hydrogel and epoxy resin were combined to extract waterlogged wooden artifacts and Marine archaeological ceramics from a 0.4-meter-deep tank. The experiment demonstrates the feasibility of using hydrogel to remove marine artifacts.
The treated subject in this manuscript is crucial and has global importance from many points of view.
The following few comments may help the readers to maximize their benefits from reading this work.
1- In the Abstract section, the authors must mention the full name of any material for the first time in the text, followed by the abbreviation. For example, PVA/TA hydrogel.
2- In section 3.1. Hydrogel Characterization: the authors need to change the colors of the columns in Figure 3e to distinguish them easily.
A minor revision is needed.
Comments on the Quality of English Language
A minor revision is needed.
Author Response
Dear reviewer.
Thank you very much for your comments and professional advice. These opinions help to improve academic rigor of our article. Based on your suggestion and request. We have made corrected modifications on the revised manuscript. All of them has been highlighted.
Responses to Reviewer #1
1 In the Abstract section, the authors must mention the full name of any material for the first time in the text, followed by the abbreviation. For example, PVA/TA hydrogel.
Answer: Thanks for your professional suggestion. The full name of any material has been modified in the text(L 12).
2 In section 3.1. Hydrogel Characterization: the authors need to change the colors of the columns in Figure 3e to distinguish them easily.
Answer: Thanks for your professional suggestion. The colors of the columns in Figure 3e has been changed. The colors of the other Figure in the text have been changed accordingly(Figure 3e, and have the same colors Figures).

Reviewer 2 Report
Comments and Suggestions for Authors
The article describes a not new, but rather original method of preservation of archaeological artifacts used during their extracting. The introduction describes the scope of the problem in exceptional detail, the methods are adequately chosen, the results and discussion reveal the essence and novelty of the developed method of temporary consolidation of marine artifacts.
A justification should be added as to why, of all the hydrogel components, only the concentration of tannic acid changes.
To improve clarity, the order of the process steps in Figure 1 should be rearranged, or the color scheme revised and the directions further highlighted.
Otherwise, the article can be published in its present form.
Author Response
Dear reviewer.
Thank you very much for your comments and professional advice. These opinions help to improve academic rigor of our article. Based on your suggestion and request. We have made corrected modifications on the revised manuscript. All of them has been highlighted.
Responses to Reviewer #2
1 A justification should be added as to why, of all the hydrogel components, only the concentration of tannic acid changes.
Answer: Thanks for your professional suggestion. The experiments had tried to changed the borax, calcium ion concentration and tannic acid concentration for testing, and it was found that the tannic acid content produces a much greater effect on the adhesion effect and mechanical strength of the hydrogel than the remaining two factors, so the tannic acid concentration was chosen as a variable to be discussed in the text.
2 To improve clarity, the order of the process steps in Figure 1 should be rearranged, or the color scheme revised and the directions further highlighted.
Answer: Thanks for your professional suggestion. The process steps in Figure 1 have been rearranged( Figure 1).

Reviewer 3 Report
Comments and Suggestions for Authors
There are some scientific interests in this manuscript. However, the quality of the paper needs to be improved. Firstly, the abbreviations of the nouns, such as PVA, TA and 3M, need to be unified, and their whole names need to be given when they first appear in the Abstract and text. Secondly, the units, such as “seconds” (page 4 line157), “minutes (page 8, line 249) and “hours” (page 11, line 313), etc. should be revised according the international requirements, leaving blanks between the numbers and units, or after brackets, such as ( ), either in the text, figures or figure legends. Thirdly, some of the expressions are not accurate and need to be improved accordingly. ······ It is better to be accepted after a major revision.

There are some scientific interests in this manuscript. However, the quality of the paper needs to be improved. Firstly, the abbreviations of the nouns, such as PVA, TA and 3M, need to be unified, and their whole names need to be given when they first appear in the Abstract and text. Secondly, the units, such as “seconds” (page 4 line157), “minutes (page 8, line 249) and “hours” (page 11, line 313), etc. should be revised according the international requirements, leaving blanks between the numbers and units, or after brackets, such as ( ), either in the text, figures or figure legends. Thirdly, some of the expressions are not accurate and need to be improved accordingly. ······ It is better to be accepted after a major revision.
Author Response
Dear reviewer.
Thank you very much for your comments and professional advice. These opinions help to improve academic rigor of our article. Based on your suggestion and request. We have made corrected modifications on the revised manuscript. All of them has been highlighted.
Responses to Reviewer #3
1 the abbreviations of the nouns, such as PVA, TA and 3M, need to be unified, and their whole names need to be given when they first appear in the Abstract and text.
Answer: Thanks for your professional suggestion. Thanks for your professional suggestion. The full name of any material has been modified in the text(L 12, L 40).
2 the units, such as “seconds” (page 4 line157), “minutes (page 8, line 249) and “hours”(page 11, line 313), etc. should be revised according the international requirements, leaving blanks between the numbers and units, or after brackets, such as ( ), either in the text, figures or figure legends.
Answer: Thanks for your professional suggestion. Blanks have been provided between numbers and units in the text in accordance with international requirements (L157, L249, L313). And blanks have been left between numbers and units in the graphic legends.(Figure 4、Figure 5、Figure 7、Figure 8、Figure 9b)
3 some of the expressions are not accurate and need to be improve accordingly.······ It is better to be accepted after a major revision.
Answer: Thanks for your professional suggestion. Some expressions in the text has been modifed.(L11-L16、L62-64、L71-74、L84-88、L215-216、L236-238、L244-246、L253-262、L267-272、L275-281、L287-288、L299、L303、L311-312、L320-332、L339-350、L361-362、L381-382、L391、L409-410).A reference [4] has been added to the line of 25.

Round 2
Reviewer 3 Report
Comments and Suggestions for Authors
The revised manuscript has been improved somewhere, but not satisfied. For example, “Guo used poly (vinyl alcohol) and poly(acrylamide) to ······” (page 2, line 87-88) should be “Guo used polyvinyl alcohol (PVA) and polyacrylamide to ······”, meanwhile the “Polyvinyl alcohol (PVA) (average Mw≈1750, 97% hydrolyzed), Tannin acid (TA) (F.W.=1701.02, >92%), Borax(F.W.=381.37, AR), CaCl2(F.W.=110.98, AR), Disodium eth-ylenediamine tetraacetic acid (Na2EDTA)(F.W.=372.24, AR) were obtained from Si-nopharm Chemical Reagent Co., Ltd(Shanghai, China)” (page 3, line 114-117) should be “PVA (average Mw ≈ 1750, 97% hydrolyzed), Tannin acid (TA) (F.W. = 1701.02, > 92%), Borax (F.W. = 381.37, AR), CaCl2 (F.W. = 110.98, AR), Disodium eth-ylenediamine tetraacetic acid (Na2EDTA) (F.W. = 372.24, AR) were obtained from Si-nopharm Chemical Reagent Co., Ltd (Shanghai, China) ”, “Upon the addition of Ca2+, it splits into two distinct peaks (1472cm-1 and 1462cm-1), suggesting the successful formation of a complexation with tannic acid [35]” (page 5, line 204-205) should be “Upon the addition of Ca2+, it splits into two distinct peaks (1472 cm-1 and 1462 cm-1), suggesting the successful formation of a complexation with TA [35]”. ······It should has a careful and thorough revision.

The revised manuscript has been improved somewhere, but not satisfied. For example, “Guo used poly (vinyl alcohol) and poly(acrylamide) to ······” (page 2, line 87-88) should be “Guo used polyvinyl alcohol (PVA) and polyacrylamide to ······”, meanwhile the “Polyvinyl alcohol (PVA) (average Mw≈1750, 97% hydrolyzed), Tannin acid (TA) (F.W.=1701.02, >92%), Borax(F.W.=381.37, AR), CaCl2(F.W.=110.98, AR), Disodium eth-ylenediamine tetraacetic acid (Na2EDTA)(F.W.=372.24, AR) were obtained from Si-nopharm Chemical Reagent Co., Ltd(Shanghai, China)” (page 3, line 114-117) should be “PVA (average Mw ≈ 1750, 97% hydrolyzed), Tannin acid (TA) (F.W. = 1701.02, > 92%), Borax (F.W. = 381.37, AR), CaCl2 (F.W. = 110.98, AR), Disodium eth-ylenediamine tetraacetic acid (Na2EDTA) (F.W. = 372.24, AR) were obtained from Si-nopharm Chemical Reagent Co., Ltd (Shanghai, China) ”, “Upon the addition of Ca2+, it splits into two distinct peaks (1472cm-1 and 1462cm-1), suggesting the successful formation of a complexation with tannic acid [35]” (page 5, line 204-205) should be “Upon the addition of Ca2+, it splits into two distinct peaks (1472 cm-1 and 1462 cm-1), suggesting the successful formation of a complexation with TA [35]”. ······It should has a careful and thorough revision.
Author Response
Dear reviewer.
Thank you very much for your comments and professional advice. These opinions help to improve academic rigor of our article. Based on your suggestion and request. We have made corrected modifications on the revised manuscript. All of them has been highlighted.
Responses to Reviewer #3
1 The revised manuscript has been improved somewhere, but not satisfied. For example, “Guo used poly (vinyl alcohol) and poly(acrylamide) to ······” (page 2, line 87-88) should be “Guo used polyvinyl alcohol (PVA) and polyacrylamide to ······”, meanwhile the “Polyvinyl alcohol (PVA) (average Mw≈1750, 97% hydrolyzed), Tannin acid (TA) (F.W.=1701.02, >92%), Borax(F.W.=381.37, AR), CaCl2(F.W.=110.98, AR), Disodium eth-ylenediamine tetraacetic acid (Na2EDTA)(F.W.=372.24, AR) were obtained from Si-nopharm Chemical Reagent Co., Ltd(Shanghai, China)” (page 3, line 114-117) should be “PVA (average Mw ≈ 1750, 97% hydrolyzed), Tannin acid (TA) (F.W. = 1701.02, > 92%), Borax (F.W. = 381.37, AR), CaCl2 (F.W. = 110.98, AR), Disodium eth-ylenediamine tetraacetic acid (Na2EDTA) (F.W. = 372.24, AR) were obtained from Si-nopharm Chemical Reagent Co., Ltd (Shanghai, China) ”, “Upon the addition of Ca2+, it splits into two distinct peaks (1472cm-1 and 1462cm-1), suggesting the successful formation of a complexation with tannic acid [35]” (page 5, line 204-205) should be “Upon the addition of Ca2+, it splits into two distinct peaks (1472 cm-1 and 1462 cm-1), suggesting the successful formation of a complexation with TA [35]”. ······It should has a careful and thorough revision.
Answer:Thanks for your professional suggestion.We have made corrected modifications on the revised manuscript. All of them has been highlighted.
